# Linking *time-series* of single-molecule experiments with molecular dynamics simulations by machine learning

**Yasuhiro Matsunaga[1,2], Yuji Sugita[1,3,4]***

[1]Computational Biophysics Research Team, RIKEN Center for Computational Science, Kobe, Japan; [2]JST PRESTO, Kawaguchi, Japan; [3]Theoretical Molecular Science Laboratory, RIKEN Cluster for Pioneering Research, Wako, Japan; [4]Laboratory for Biomolecular Function Simulation, RIKEN Center for Biosystems Dynamics Research, Kobe, Japan

**Abstract** Single-molecule experiments and molecular dynamics (MD) simulations are indispensable tools for investigating protein conformational dynamics. The former provide *time-series* data, such as donor-acceptor distances, whereas the latter give atomistic information, although this information is often biased by model parameters. Here, we devise a machine-learning method to combine the complementary information from the two approaches and construct a consistent model of conformational dynamics. It is applied to the folding dynamics of the formin-binding protein WW domain. MD simulations over 400 μs led to an initial Markov state model (MSM), which was then "refined" using single-molecule Förster resonance energy transfer (FRET) data through hidden Markov modeling. The refined or *data-assimilated* MSM reproduces the FRET data and features hairpin one in the transition-state ensemble, consistent with mutation experiments. The folding pathway in the data-assimilated MSM suggests interplay between hydrophobic contacts and turn formation. Our method provides a general framework for investigating conformational transitions in other proteins.
DOI: https://doi.org/10.7554/eLife.32668.001

**\*For correspondence:**
sugita@riken.jp

**Competing interests:** The authors declare that no competing interests exist.

## Introduction

Protein folding is an important subject not only for basic research in molecular biology but also for understanding folding diseases and designing new polymeric materials (*Dill and MacCallum, 2012*). Transient, partially folded states are often encountered on folding pathways, and have been characterized experimentally in solution by methods such as laser temperature jumps, fluorescence labeling, and solution X-ray scattering. Mutagenesis evaluated by Φ-value analysis (*Fersht and Daggett, 2002*), for instance, has also provided residue-level information on transition states. Recently, state-of-the-art single-molecule (sm) measurements, single-molecule Förster resonance energy transfer (smFRET) (*Chung et al., 2012*; *Oikawa et al., 2013*) and force spectroscopy (smFS) (*Neupane et al., 2016*) have become powerful tools in protein-folding research, providing reliable information on transition-path (barrier-crossing process) times (*Chung et al., 2012*) and heterogeneity in the unfolded state (*Oikawa et al., 2013*). A major limitation of smFRET is that the observables are restricted to 'low-dimensional' structural data, such as the donor–acceptor distance. Computational modeling should help us to interpret single-molecule *time-series* measurements and should contribute to solving some remaining puzzles, such as the reduced solvent viscosity dependence of the transition-path times (*Chung and Eaton, 2013*), the internal viscosity (*Chung and Eaton, 2013*) and the impact of non-Markovian property (*Chung et al., 2015*). Theories and

computational methods have been developed to extract structural information and dynamics from smFRET data (*Haas et al., 2013*; *Hoefling et al., 2011*; *Matsunaga et al., 2015*; *Sun et al., 2016*).

Molecular dynamics (MD) simulation is another powerful approach for investigating protein dynamics and folding over relatively long time periods — hundreds of microseconds or longer (*Lindorff-Larsen et al., 2011*). In theory, 'low-dimensional' smFRET measurements are interpreted in terms of the atomic structural models obtained with MD simulations. However, it is still a challenge to achieve quantitative agreement between simulation and experimental data during the entire folding process, owing not only to simulation time limitations but also to inherent force-field biases. In particular, while local interactions are well described by the current force fields, it is still difficult to reproduce energetic balances between unfolded and folded states. Indeed, Piana and coworkers showed, in their protein folding simulation study, that the folding mechanism of the villin headpiece depends substantially on the choice of force field (*Piana et al., 2011*). It is also known that most force fields produce unfolded states that are more compact and structured than those suggested experimentally (*Piana et al., 2014*). Methods based on maximum entropy (*Beauchamp et al., 2014*; *Boomsma et al., 2014*; *Cavalli et al., 2013*; *Olsson et al., 2013, 2014, 2017*; *Pitera and Chodera, 2012*; *Roux and Weare, 2013*) or Bayesian statistics (*Bonomi et al., 2016*) were recently developed for guiding simulations or models to generate ensembles that match experimental *ensemble-averaged observations*.

Exploiting *time-series* data from single-molecule experiments is another way to link simulation with experiment and has several advantages over the ensemble-average based approaches: (i) more latent states can be uncovered by inferring states from their historical evolution than from their static ensembles (*Li et al., 2008*; *Matsunaga et al., 2015*; *Schuetz et al., 2010*); (ii) the transition state can be uniquely identified as a dynamic bottleneck by following the actual dynamics. However, the time-scale gap between experiment and simulation previously hampered the direct use of *time-series* analysis methods in other disciplines, such as data assimilations that are based on the sequential Monte Carlo method (*Matsunaga et al., 2015*).

Here, we develop a new approach for single-molecule *time-series* data based on the Markov state model (MSM) (*Pande et al., 2010*), a statistical model that approximates dynamics by memory-less probabilistic transitions between discrete conformational states. In MSM, the probability of transition from the discrete state $i$ to state $j$ in a lag time of $\tau$ is described by a transition-probability matrix $\mathbf{T}(\tau) = \left\{ T_{ij}(\tau) \right\}$. $\mathbf{T}(\tau)$ can be estimated from a set of short simulations instead of a single long simulation. Therefore, MSM is often used to generate long-time dynamics (*Silva et al., 2014*) whose time-scale is comparable to those of experimental measurements (*Feng et al., 2016*; *Noé et al., 2011*; *Snow et al., 2002*). However, estimation of $\mathbf{T}(\tau)$ is largely dependent on the simulation force-fields, which may have uncertainties or biases for certain conformations (*Olsson et al., 2017*). To overcome this problem, we propose to "refine" $\mathbf{T}(\tau)$ using high-resolution measurement of single-molecules as *time-series* data. Specifically, we use a machine-learning approach to estimate $\mathbf{T}(\tau)$ between hidden Markov states from low-dimensional *time-series* data (*Bishop, 2006*). To distinguish the original $\mathbf{T}(\tau)(= \mathbf{T}_{\text{simulation}}(\tau))$, we refer to the refined $\mathbf{T}(\tau)$ as $\mathbf{T}_{\text{experiment}}(\tau)$ in this paper.

We propose a two-step procedure in our machine-learning approach, which links simulations and single-molecule experiments (*Figure 1*). (i) *Supervised learning step*. We first construct an initial MSM from a raw set of simulation data. After defining discrete conformational states by clustering the trajectory snapshots, $\mathbf{T}(\tau) = \mathbf{T}_{\text{simulation}}(\tau)$ is estimated directly by counting transitions between the discrete states in the simulation trajectories. This step is the same as conventional MSM (*Pande et al., 2010*). (ii) *Unsupervised learning step*. Using $\mathbf{T}_{\text{simulation}}(\tau)$ as an initial estimate, we perform hidden Markov modeling (*Bronson et al., 2009*; *McKinney et al., 2006*; *Okamoto and Sako, 2012*; *Pirchi et al., 2016*; *Rabiner and Juang, 1986*; *Schröder and Grubmüller, 2003*) to refine the initial MSM using single-molecule measurement *time-series* data. $\mathbf{T}(\tau)$ is optimized so that the "refined" or *data-assimilated* MSM with $\mathbf{T}_{\text{experiment}}(\tau)$ can reproduce the *time-series data* most accurately.

We applied this procedure to the folding dynamics of the formin-binding protein (FBP) WW domain, a 37-residue three-stranded β-sheet protein. In the construction of the initial MSM, extensive MD simulations of a dye-labeled WW domain were performed for an aggregated time of ~400 μs. Although there are a number of folding simulation studies for the FBP WW domain and its homologs (*Ensign and Pande, 2009*; *Freddolino et al., 2008*; *Karanicolas and Brooks, 2003*; *Zanetti-*

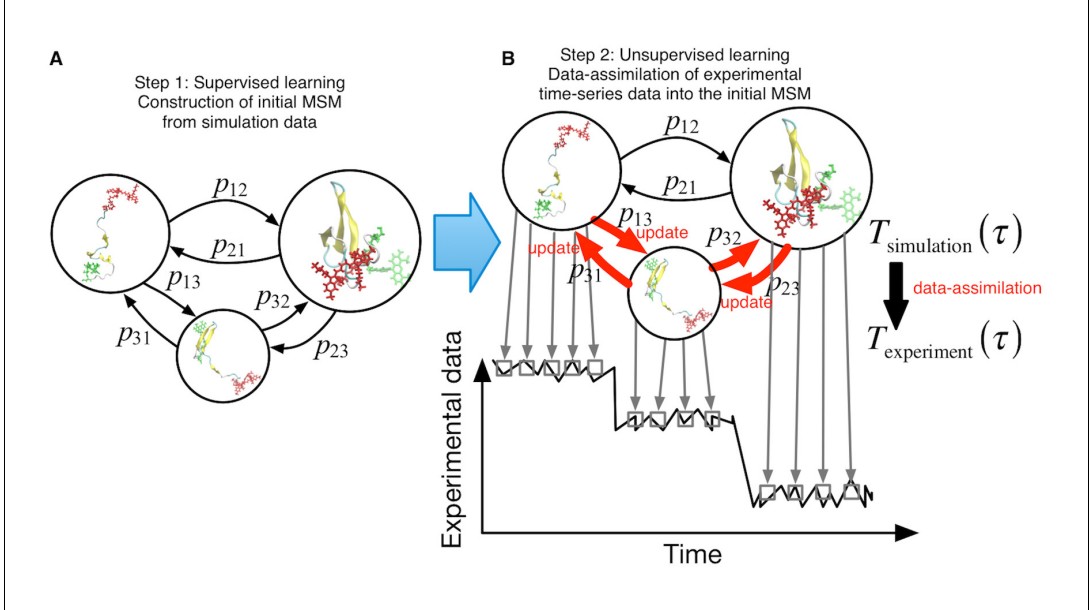

**Figure 1.** Schematic of proposed semi-supervised learning approach. (**A**) Our proposed approach comprises two steps. As the first step, an initial Markov State Model is constructed only from simulation data by simply counting transitions between conformational states. (**B**) In the second step, transition probabilities (depicted by arrows) are updated through unsupervised learning from experimental time-series data.

DOI: https://doi.org/10.7554/eLife.32668.002

The following figure supplement is available for figure 1:

**Figure supplement 1.** Dye-labeled WW domain and simulation box.

DOI: https://doi.org/10.7554/eLife.32668.003

*Polzi et al., 2017*; *Zhou et al., 2014*), this may be the first folding simulation to use FRET dyes. High time-resolution smFRET measurements of WW domain folding and unfolding dynamics were used for the unsupervised learning (*Chung et al., 2012*). The data were previously measured to resolve the durations of the folding and unfolding transitions with microsecond resolution by Chung and coworkers (*Chung et al., 2012*). The initial and data-assimilated MSM showed different folding pathways and transition-state ensembles of WW domains. An independent mutation experiment with Φ-value analysis (*Fersht and Daggett, 2002*) validated the data-assimilated MSM. We discuss our time-series analysis method in the context of machine-learning theory and its applicability to conformational transitions in other biomolecules.

## Results

### Single-molecule FRET measurements

SmFRET experiments were carried out by Chung and coworkers, and details for the experiments are given in *Chung et al., 2012*. Here we summarize the essential experimental setups necessary for our machine-learning procedure. Photon trajectories were measured for the FBP WW domain with donor (Alexa 488) and acceptor (Alexa 594) fluorophores attached to the terminal residues in the protein. In order to improve the time resolution of the smFRET data, Chung and coworkers illuminated the protein with a very high intensity laser (10 kW/cm$^2$), increasing the number of photons observed per time (~650 photons/ms) (*Chung et al., 2012*). Photon color, either donor green or acceptor red, and the absolute time of arrivals were recorded within ~0.5 ns. Each photon trajectory was split into folded and unfolded segments by finding the photon interval with the maximum transition probability (*Gopich and Szabo, 2009*). The final set of smFRET photon sequences comprises 527 trajectories, each of which contains a single folding or unfolding event.

## Molecular dynamics simulations

A dye-labeled WW domain was built in silico for MD simulations (*Figure 1—figure supplement 1*). Starting from the NMR structure of the WW domain (PDB code: 1E0L [*Macias et al., 2000*]), we made the same substitution mutation W30A as in the experiments (*Chung et al., 2012*), and the terminal residues were labeled with donor (Alexa Fluor 488) or acceptor (Alexa Fluor 594) dyes by using the AMBER-DYES package (*Graen et al., 2014*).

We conducted eleven simulations of length 25.6 µs in the NVT ensemble (370 K) from the unfolded structures. By monitoring the fraction of native contacts, $Q$, we observed folding events in four trajectories out of the eleven (*Figure 2—figure supplement 1A*). Also, six simulations of length 10 µs and four simulations of 14 µs were performed from the folded structure. In these ten trajectories, we observed seven unfolding events. Our simulations, thus, sampled intermediate regions between the unfolded and folded states sufficiently for construction of an initial MSM. The simulation length is ~400 µs in total.

## Clustering of sampled structures in MD simulations

For the MSM construction, we chose a two-dimensional (2-D) space spanned by the native contact $Q$ and the expected FRET efficiency ε. The value of ε is calculated from the distance $r$ between donor and acceptor dyes using the Förster theory, $\varepsilon = 1/\left[1 + (r/R_0)^6\right]$, where $R_0$ is the Förster radius (see 'Methods' for details). Here, $Q$ is chosen because it is historically known to be the best reaction coordinate to describe a folding process. Also, ε is employed for comparison with smFRET data as well as for differentiating compact and elongated structures. *Figure 2A* shows a scatter plot of the sampled conformations in the MD simulations. Expected FRET efficiency ε successfully resolves the elongated unfolded states and compact states, whereas ε fails to discriminate the folded state (corresponding to $Q \sim 0.7$–1.0 and $\varepsilon \sim 0.7$–1.0) from the compact unfolded state ($Q \sim 0.0$–0.3 and $\varepsilon \sim 0.7$–1.0) without the help of $Q$. This suggests that approaches that are based on ensemble average may have a difficulty when the histogram of ε is used to add biases to the protein in MD simulations.

Sampled conformations were clustered into discrete states in the 2-D space (cluster centers are shown in *Figure 2B*). Regular spatial clustering was applied to partition the space in an equidistant manner regardless of the population size (*Senne et al., 2012*). This spatial clustering is essential in this study because relatively minor populations can have high probabilities after the refinement of the initial MSM with the help of experimental *time-series* data.

In order to obtain structural insights, we calculated both the mean and the variance of the donor-acceptor distance $r$ for each state. *Figure 2—figure supplement 2* shows samples plotted in the 2D space spanned by $Q$ and $r$. There are distance gaps between states, which are supportive of the Markovian assumption. Large standard deviations are observed in the case of small donor-acceptor distances. This results from the lower spatial resolution in FRET efficiency ε when ε is not close to 0.5 (e.g., states with $\varepsilon \sim 1$ cover $r = 0$-30 Å) as discussed in the information-binning study by *Watkins and Yang (2004)*.

## Construction of an initial MSM as the supervised learning step

MSM has two tunable parameters: the number of discrete states and the lag time τ for $\mathbf{T}(\tau)$. Theoretically, increasing the number of discrete states and/or the lag time will produce a MSM with the least discretization error (*Prinz et al., 2011*), while a large number of discrete states or a longer lag time will decrease the number of samples for estimation, resulting in large statistical errors (the so-called bias-variance tradeoff) (*McGibbon et al., 2014*). Thus, it is common to make the number of discrete states and the lag time as small as possible, even though it compromises the accuracy of the model. Here, we examined various numbers of discrete states for MSM by calculating implied time scales (*Schwantes and Pande, 2013*). The *i*th implied time scale $t_i$ of a MSM with $\mathbf{T}(\tau)$ is given by

$$t_i = -\frac{\tau}{\ln \lambda_i}, \tag{1}$$

where $\lambda_i$ is the *i*th eigenvalue of $\mathbf{T}(\tau)$. As the implied time scales are always underestimated relative to their true values, the slower time scales are indicative of smaller discretization errors. The

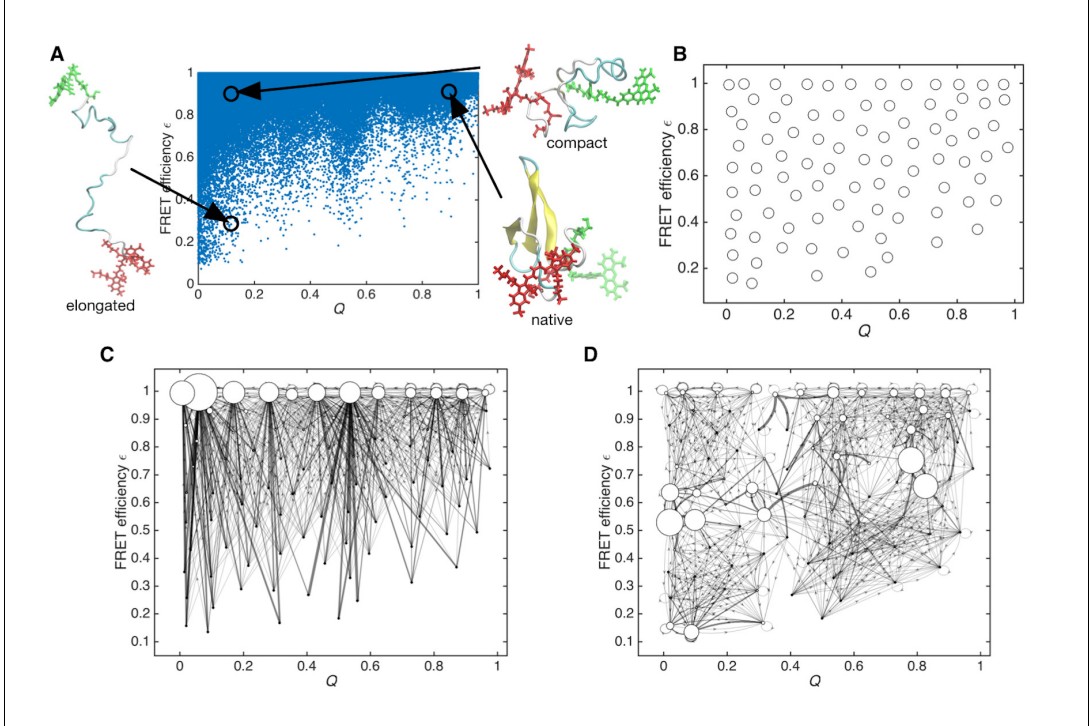

**Figure 2.** Sampled conformations from simulations and Markov state models constructed in $Q$ and expected FRET efficiency space. (**A**) Scatter plot of sampled conformations from the aggregated trajectories. Representative structures from folded, compact unfolded, and elongated states are shown. Donor and acceptor dyes are colored green and red, respectively. (**B**) Cluster centers used for constructing the Markov state model are plotted with circles. (**C**) Initial Markov state model constructed from simulation data only. Node areas are proportional to the equilibrium populations, and edge line widths are proportional to the transition probabilities. (**D**) Data-assimilated Markov state model after unsupervised learning from smFRET photon-count sequences. Edges with transition probabilities of less than 0.01 are not shown for visual clarity.

DOI: https://doi.org/10.7554/eLife.32668.004

The following figure supplements are available for figure 2:

**Figure supplement 1.** $Q$ of molecular dynamics simulation trajectories.
DOI: https://doi.org/10.7554/eLife.32668.005

**Figure supplement 2.** Donor-acceptor distances of the Markov states.
DOI: https://doi.org/10.7554/eLife.32668.006

**Figure supplement 3.** Implied timescales for various numbers of states.
DOI: https://doi.org/10.7554/eLife.32668.007

**Figure supplement 4.** Comparison of the transition probabilities of the initial and the data-assimilated Markov state models.
DOI: https://doi.org/10.7554/eLife.32668.008

**Figure supplement 5.** Data-assimilated Markov state models using halves of the training data.
DOI: https://doi.org/10.7554/eLife.32668.009

**Figure supplement 6.** Dependency of data-assimilated Markov state models on the choice of Förster radius $R_0$.
DOI: https://doi.org/10.7554/eLife.32668.010

**Figure supplement 7.** Data-assimilated Markov state obtained by considering the FRET efficiency outside the weak-excitation limit.
DOI: https://doi.org/10.7554/eLife.32668.011

**Figure supplement 8.** Optimization process for the initial Markov state model.
DOI: https://doi.org/10.7554/eLife.32668.012

**Figure supplement 9.** Optimization of a random matrix as the initial condition.
DOI: https://doi.org/10.7554/eLife.32668.013

implied time scales were calculated for various numbers of clustered states as a function of the lag time $\tau$ (*Figure 2—figure supplement 3A*). We found that the converged values of the slowest implied time scale (related to folding dynamics) successfully reproduce the time scale of folding (~5 µs) in the MD data when the number of states is larger than 80 (*Figure 2—figure supplement 3B*).

From these observations, we chose 87 states by adjusting the cluster radius (0.08) in the regular spatial clustering.

A lag time of $\tau = 200$ ns was chosen as a minimum time scale to achieve converged implied time scales. Note that the number of states chosen here is an order of magnitude fewer than those in other MSM studies. It is well known that the RMSD metric requires a larger number of discrete states in MSM, whereas well-defined metrics that are based on slow motions or smooth coordinates (such as contact-maps [*Kellogg et al., 2012*], or coordinates extracted by time-structure-based independent component analysis [*Schwantes and Pande, 2013*]) require a smaller number of states. A long lag time of $\tau = 200$ ns also helps the MSM to satisfy the Markov assumption as well as improving FRET photon-count statistics in the next unsupervised learning step.

*Figure 2C* shows a graphical representation of the initial MSM constructed only from MD simulation data. The node areas are proportional to the equilibrium populations, and the edge line widths are proportional to the transition probabilities, $T_{ij}(\tau)$. The MSM seems to overemphasize compact unfolded states ($Q \sim 0.0$–$0.3$ and $\varepsilon \sim 0.7$–$1.0$), which results from biases of the force-field parameters not only for proteins (*Best et al., 2014*; *Piana et al., 2014*) but also for FRET dyes (*Best et al., 2015*).

## Refinement of transition probabilities as the unsupervised learning step

In the unsupervised learning, the total log likelihood function of all smFRET photon-counting sequences $\ln L(\mathbf{T}(\tau)) = \sum_k \ln L_k(\mathbf{T}(\tau))$ is maximized by optimizing $\mathbf{T}(\tau)$. First, we treated smFRET data as $N$ independent photon-counting sequences in discretized time windows. Each sequence $O_k = \left\{ o_1^{(k)} \ldots o_I^{(k)} \right\}$ consists of a set of donor and acceptor photon counts $o_i = \left\{ N_D^{(i)}, N_A^{(i)} \right\}$ detected in $i$th time window. $I$ is the number of time windows and 200 ns was chosen for the photon-counting time-window width as well as for the lag time $\tau$ of MSM. The likelihood function $L_k(\mathbf{T}(\tau))$ is then defined as a probability to observe the $k$th smFRET photon-counting sequence $O_k$ with a given $\mathbf{T}(\tau)$ (*Gopich and Szabo, 2012*):

$$L_k(\mathbf{T}(\tau)) = p(O_k|(\mathbf{T}(\tau))) = \sum_{s_1=1}^{M} \cdots \sum_{s_I=1}^{M} p(s_1)h(o_1|s_1) \prod_{i=2}^{I} p(s_i|s_{i-1})h(o_i|s_i). \qquad (2)$$

$M$ is the number of discrete states in MSM, and $s_i$ denotes MSM's state at the $i$th time window. $p(s_i|s_{i-1}) = T_{s_{i-1}s_i}(\tau)$ is the transition probability from state $s_{i-1}$ to state $s_i$. $p(s_1)$ is the equilibrium probability of being in state $s_1$. $h(o_i|s_i)$ is the probability of observing donor and acceptor photon-counts $o_i = \left\{ N_D^{(i)}, N_A^{(i)} \right\}$ given state $s_i$.

By maximizing the above likelihood function, we obtained the data-assimilated MSM with the optimized $\mathbf{T}(\tau) = \mathbf{T}_{\text{experiment}}(\tau)$, which matches with the smFRET *time-series* data. *Figure 2D* shows that the data-assimilated MSM differs from the initial MSM. In the data-assimilated MSM, the compact unfolded state ($Q \sim 0.0$–$0.3$ and $\varepsilon \sim 0.7$–$1.0$) disappears due to a minor population with high $\varepsilon$ in the smFRET data (see *Figure 3B*), while an elongated unfolded region ($Q \sim 0.0$–$0.2$ and $\varepsilon \sim 0.5$–$0.6$) is stabilized. As noted, this may reflect the biases of the force-field. Also, the different solvent conditions between the simulation (TIP3P water molecules) and the smFRET experiment (denaturant concentration) may affect the unfolded state distribution (*Zheng et al., 2016*). Interestingly, as another stable region, the folded state ($Q \sim 0.8$ and $\varepsilon \sim 0.6$–$0.8$) appears instead of other states with the same $\varepsilon$.

We compared the transition probabilities $T_{ij}(\tau)$ of the two MSMs (*Figure 2—figure supplement 4*). *Figure 2—figure supplement 4A* shows the implied time scales of the MSMs. We can see that the slowest time scale increases from 2.6 μs to 100 μs after the hidden Markov modeling. These time scales are related to folding/unfolding transitions of WW domain in both cases. Simulation time scale is faster than that in experiments because of the lower viscosity of the TIP3P water model (*Mahoney and Jorgensen, 2001*) compared to those of pure water and the viscogen added in the smFRET experiment (*Chung et al., 2012*). This gap was improved by the information from the smFRET data. In *Figure 2—figure supplement 4B*, $T_{ij}(\tau)$ are directly compared in a scatter plot where each $T_{ij}(\tau)$ is colored using the FRET efficiency $\varepsilon$ of state $i$ before transition. $T_{ij}(\tau)$ from states with high FRET efficiencies are correlated with each other. This means that $T_{ij}(\tau)$ related to compact

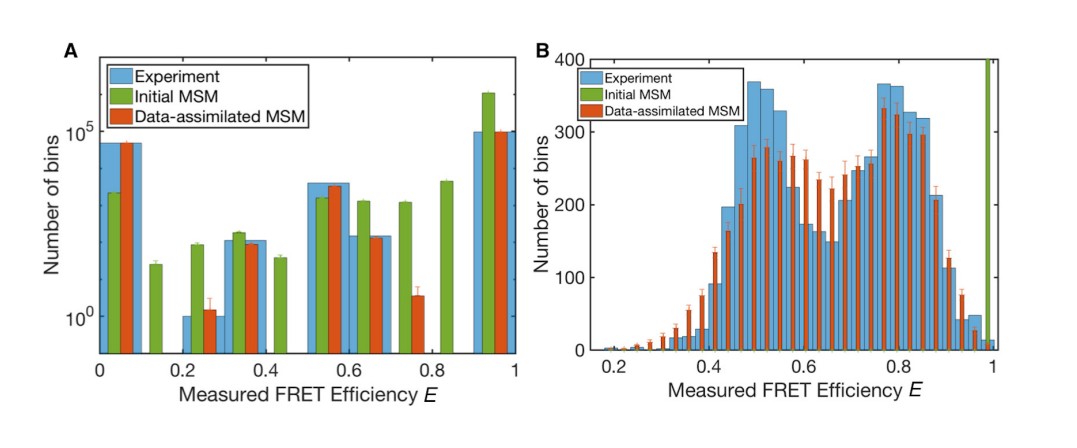

**Figure 3.** Measured FRET efficiency histograms. (A) Measured FRET efficiency histograms calculated from donor and acceptor photons in the single-molecule FRET data with a time-window of 200 ns width, and those generated from initial and data-assimilated Markov state models. The measured FRET efficiency is defined as the ratio of the acceptor photon counts to the total number of photons ($E = N_A/(N_A + N_D)$). Error bars indicate standard deviations in ten realizations of photon sequences for both models. (B) Measured FRET efficiency histograms calculated with a time-window of 50 μs.

DOI: https://doi.org/10.7554/eLife.32668.014

The following figure supplement is available for figure 3:

**Figure supplement 1.** *k*-fold cross validation test.

DOI: https://doi.org/10.7554/eLife.32668.015

states derived from MD simulations are consistent with the smFRET data. By contrast, $T_{ij}(\tau)$ from states with middle or low FRET efficiencies are less correlated. This means that the hidden Markov modeling mainly updated $T_{ij}(\tau)$ of elongated states.

## Reproducibility of experimental data

*Figure 3* shows histograms of the 'measured' FRET efficiency $E$ of the original smFRET data and those generated by the initial and data-assimilated MSMs (see 'Methods' for emulation or stochastic simulation of smFRET data). The measured FRET efficiency $E$ is calculated from the numbers of photons emitted from donor and acceptor dyes in a certain time window, $N_D$ and $N_A$, respectively. It is defined by the ratio of donor photon counts to total photon counts, $E = N_A/(N_A + N_D)$. $E$ is calculated from measured photons in a time window whereas ε is calculated from the donor-acceptor distance of each instantaneous structure. In *Figures 3A*, 200 ns time-windows were used for photon counts. The data-assimilated MSM produced a histogram close to that found with original smFRET data (the mean squared error between the normalized histograms is $3.7 \times 10^{-5}$), confirming the reliability of the optimized parameters obtained by machine learning.

We also calculated the observed FRET efficiency $E$ with 50 μs time windows (*Figure 3B*), which corresponds to the folding time-scale observed in the smFRET data (*Chung et al., 2012*). The histogram of the original smFRET data has double peaks, corresponding to unfolded and folded states, respectively. The initial MSM, however, only shows a single sharp peak at high FRET efficiency because of overemphasis of compact conformations (*Best et al., 2014*). The data-assimilated MSM successfully reproduces the double peaks of the histogram of the original data. The histogram of the data-assimilated MSM seems smoothed compared to that obtained experimentally, presumably because of the accumulation of photon-counting noise using 200 ns time-windows.

## Folding mechanisms of the FBP WW domain

To quantify the difference between the initial and data-assimilated MSMs, we calculated $p_{fold}$ by applying transition-path theory (*Metzner et al., 2009*; *Noé et al., 2009*) (*Figure 4A and B*). $p_{fold}$ is the probability of undergoing a folding transition defined for each state. States with $p_{fold} > 0.5$ are kinetically closer to the folded state, whereas those with $p_{fold} < 0.5$ approximate the unfolded state. Those with $p_{fold} = 0.5$ define the transition-state ensemble. The calculated $p_{fold}$ tends to depend

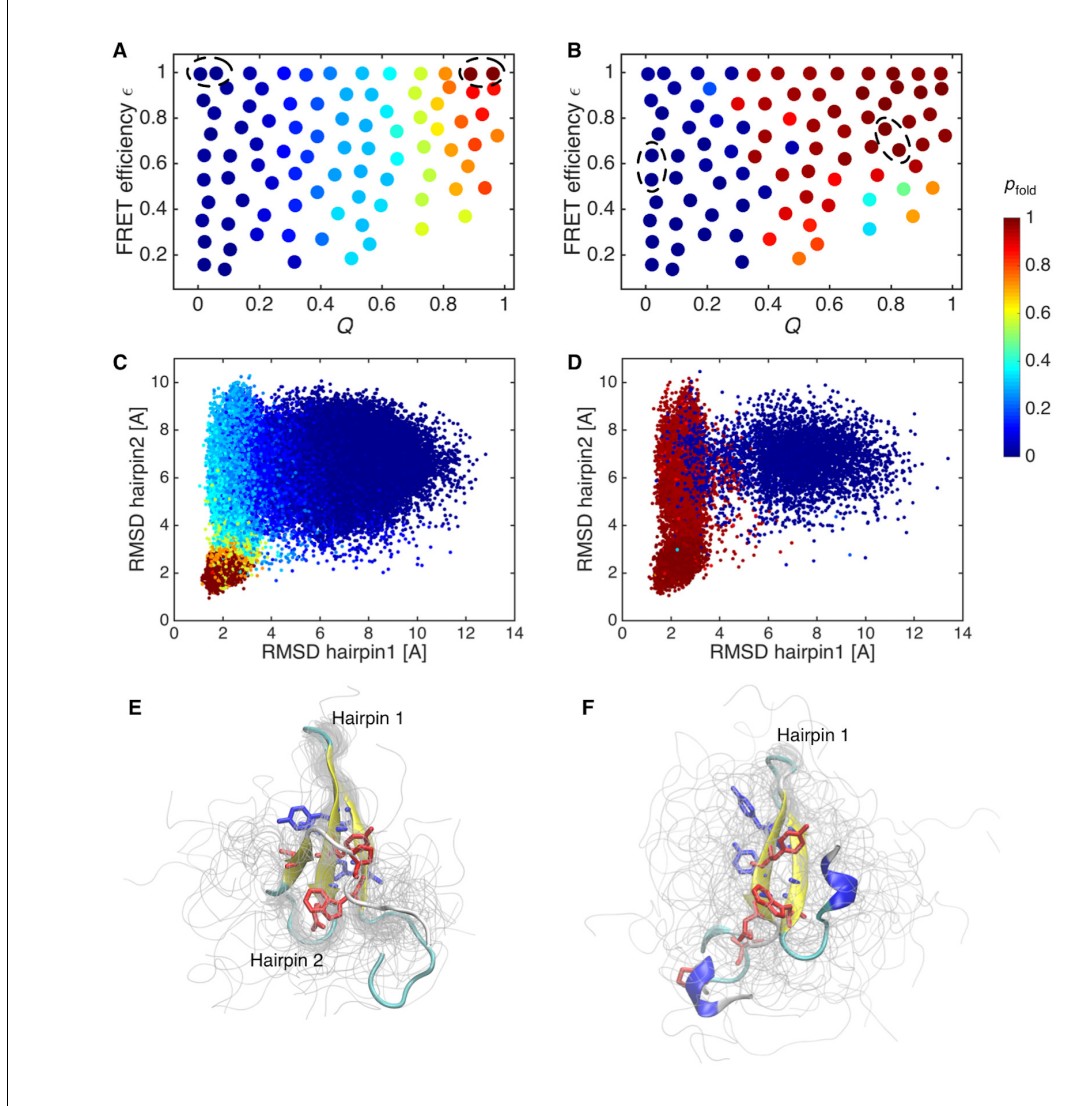

**Figure 4.** Probability of folding, $p_{fold}$, and transition state ensemble. (A) Probabilities of folding, $p_{fold}$, mapped onto the states of initial Markov state model. (B) $p_{fold}$ for data-assimilated Markov state model. The unfolded and folded states used for the calculation (source and sink in the context of the transition path theory, respectively) are indicated by circles. (C) Trajectory snapshots in the the RMSDs of hairpins 1 and 2 from their native structures are colored by $p_{fold}$ for the initial Markov state model. (D) Trajectory snapshots of the data-assimilated Markov state model. (E) Structures of the transition state ensemble in the initial Markov state model which correspond to $p_{fold}$ = 0.4–0.6. (F) The transition state ensemble in the data-assimilated Markov state model. Two hydrophobic cores that project below and above the plane of the sheet, core 1 (Trp8, Tyr20, Asn22, Thr29, Pro33, shown in red) and core 2 (Thr9, Tyr11, Tyr 19, Tyr21, shown in blue) are represented by sticks.
DOI: https://doi.org/10.7554/eLife.32668.016

The following figure supplement is available for figure 4:

**Figure supplement 1.** Dynamics of initial and data-assimilated Markov state models.
DOI: https://doi.org/10.7554/eLife.32668.017

only on $Q$ for the initial MSM (*Figure 4A*), as corroborated by a previous analysis of folding simulation data for various proteins (*Best et al., 2013*). On the other hand, $p_{fold}$ of the data-assimilated MSM depends on $\epsilon$ as well as $Q$, suggesting that not only $Q$ but also compactness needs to be factored into the folding mechanism (*Figure 4B*). $p_{fold}$ was mapped onto the RMSDs of hairpins 1 and 2 of the native structure (*Figure 4C and d*). In the initial MSM, the transition-state region ($p_{fold}$ = 0.4–0.6) is located in a rather compact region where the formation of both hairpins 1 and 2 can be just discerned (*Figure 4E*). In the data-assimilated MSM, the transition-state ensemble presents only

hairpin 1 (*Figure 4F*). This is consistent with a mutagenesis experiment, where mutations in hairpin 1 produce large Φ-values (*Petrovich et al., 2006*), implying that formation of hairpin 1 contributes energetically to the transition-state ensemble.

The flux of folding trajectories can be decomposed into individual pathways for both models (*Figure 5A and B*). The decomposition extracts a set of pathways along with their fluxes. The dominant pathways with large fluxes provide the statistically probable order of events during folding. The figures show that folding pathways with largest fluxes contribute 50% of the total flux. In the data-assimilated MSM (*Figure 5B*), the formation of hydrophobic side-chain cores (core 1 consists of Trp8, Tyr20, Asn22, Thr29 and Pro33, and core 2 consists of Thr9, Tyr11, Tyr 19 and Tyr21) stabilizes the β-sheet structure in hairpin 1. The increased stability of hairpin 1 seems to guide the formation of the second hairpin (hairpin 2) by the inter-strand hydrophobic interactions. Again, this scenario is consistent with site-directed mutagenesis experiments for FBP (*Petrovich et al., 2006*) and the homologous Pin WW domains (*Jäger et al., 2001*). These experiments implied that interactions

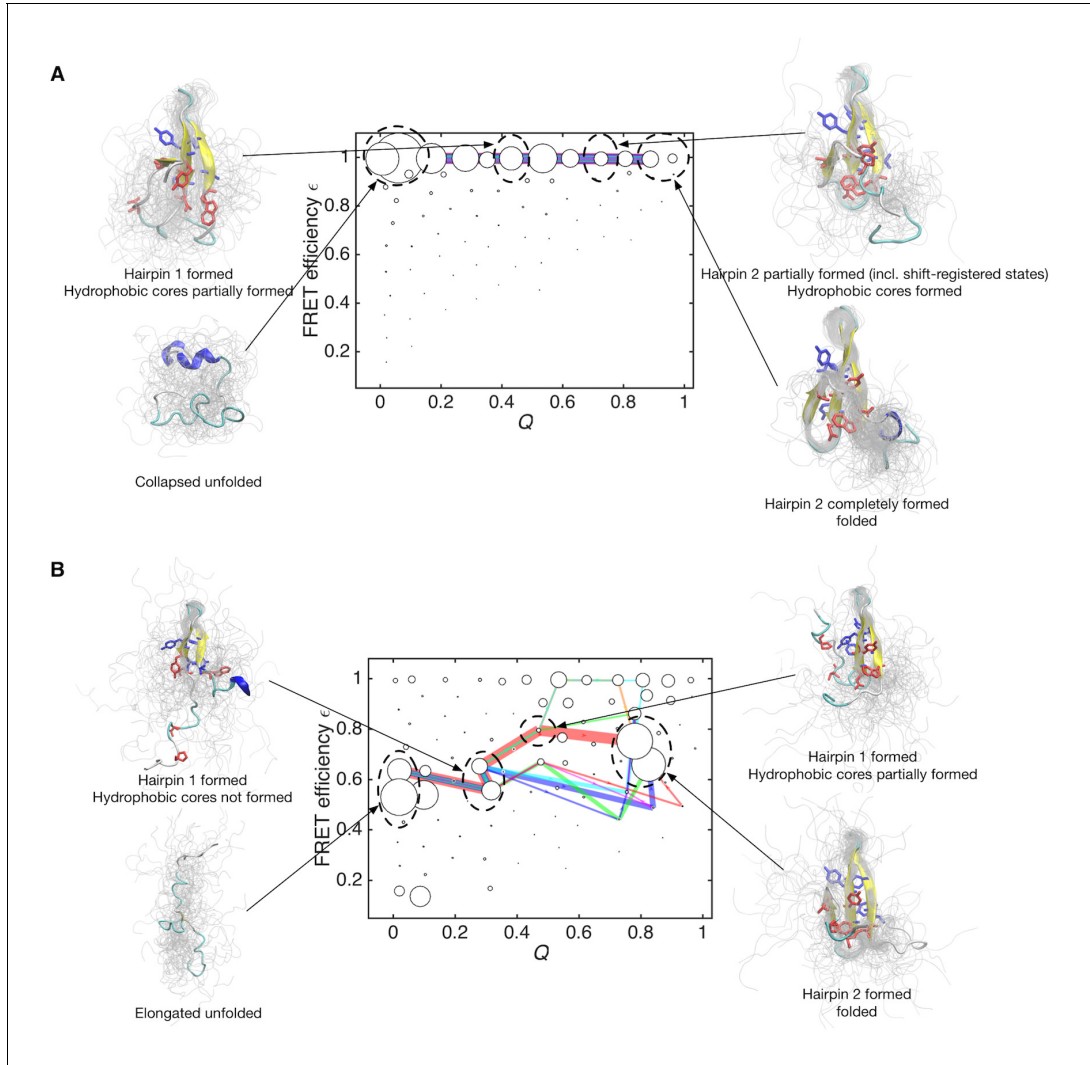

**Figure 5.** Folding pathways for initial and data-assimilated Markov state models. (**A**) Folding flux of the initial Markov state model was decomposed into individual folding pathways. Folding pathways with largest fluxes contributing 50% of the total flux are superimposed in different colors. In this case, all of the pathways are located at expected FRET efficiency ε ~ 1 with different step size in the Q axis. Line widths are proportional to fluxes. Structures of representative states are shown. Two hydrophobic cores that project below and above the plane of the sheet, core 1 (Trp8, Tyr20, Asn22, Thr29 and Pro33, shown in red) and core 2 (Thr9, Tyr11, Tyr 19 and Tyr21, shown in blue) are represented by sticks. (**B**) Folding pathways with largest fluxes contributing 50% of the total flux are shown for the data-assimilated Markov state model.

DOI: https://doi.org/10.7554/eLife.32668.018

between conserved hydrophobic residues contribute to the stability of only the native state and not to the transition state ensemble. Previous simulation studies have suggested the existence of register-shifted structures as trapped (*NoeNoé et al., 2009*) or intermediate (*Mu et al., 2006*) states, whereas in the data-assimilated MSM, such states were rarely observed. Hairpin 1 formation as a rate-limiting step together with the paucity of register-shifted states in the data-assimilated MSM are consistent with the scenario of the Wako-Saitô-Muñoz-Eaton (WSME) model (*Muñoz et al., 1997*; *Wako and Saitô, 1978*), which postulates that the formation of a local turn is a bottleneck for β-sheet formation. Interestingly, hairpin 2 formation is driven by a hydrophobic collapse (*Dinner et al., 1999*) in the data-assimilated MSM. This implies that the interplay between turns and hydrophobic clusters has an important role in the formation of multiple hairpins in β-sheet proteins.

In terms of the theory of the coil-globule transition (collapse transition) (*Ziv et al., 2009*), the formation of hairpin 1 would be the collapse step. This collapse transition is specific in the sense that the collapsed structure does not contain a large number of non-native contacts, whereas the collapse of homopolymers are often treated as a non-specific transition. As predicted by the theory (*Ziv et al., 2009*), this indicates that the folding and collapse transition temperatures are close for this WW domain.

## Discussion

We have proposed a two-step procedure for the construction of a data-assimilated MSM with $\mathbf{T}(\tau) = \mathbf{T}_{\text{experiment}}(\tau)$ matching single-molecule *time-series data*. Using smFRET data for the FBP WW domain, we show that the data-assimilated MSM successfully reproduces the original smFRET data, and yields a transition-state ensemble consistent with an independent mutational experiment (*Petrovich et al., 2006*). The folding mechanism based on the data-assimilated MSM suggests an interplay between hairpin and hydrophobic formations.

In the context of machine-learning theory, the proposed two-step procedure can be regarded as a semi-supervised learning algorithm, which tries to learn from both *labeled* and *unlabeled* data (*Rudzinski et al., 2016*; *Zhu and Goldberg, 2009*). In the context of MSM, simulation data correspond to labeled data while experimental data are unlabeled data. In a typical case of semi-supervised learning (e.g., image recognition), the labeled data are correct and usually expensive (e.g., images that are manually labeled by investigator). Therefore, unlabeled data are often 'de-emphasized' by scaling their contribution in the likelihood function (*Zhu and Goldberg, 2009*). By contrast, in our case, labeled data (simulation) may have incorrect transition counts caused by force-field biases whereas unlabeled data (experimental) possess more reliable information on dynamics. Thus, in our two-step procedure, the estimates $\mathbf{T}_{\text{simulation}}(\tau)$ from labeled data (simulation) are replaced with $\mathbf{T}_{\text{experiment}}(\tau)$ "refined" with unlabeled data (experiment). This is regarded as the limit of 'de-emphasis' on labeled data (simulation).

When fitting a rather complex model to any experimental data, the model can overreact to noise in the data (the overfitting problem). In particular, our MSM for the FBP WW domain has a rather large number of parameters ($87 \times 87$ transition probabilities), which could be easily overfitted to the smFRET data. To assess the overfitting in the unsupervised learning, we divided the smFRET data in half, and unsupervised learning was independently applied to the two subsets. Qualitatively similar network structures appeared in both, and were similar to that obtained with the full data set (*Figure 2—figure supplement 4*). This is because the effective number of parameters is considerably reduced from $87 \times 87$ down to only those involving populated states ($\varepsilon \sim 0.5$–$0.8$). In order to counteract overfitting, a maximum caliber approach for minimally perturbing the initial MSM could be a promising direction for a future study (*Dixit and Dill, 2014, 2015, 2018*; *Wan et al., 2016*; *Zhou et al., 2017*). Furthermore, to see the dependence on the choice of the Förster radius $R_0$, we carried out the unsupervised learning using a set of different $R_0$ values ($R_0 = 54, 55, 57$, and $58$ Å). The overall structure of the MSM network was robust against the choice of $R_0$ except for $R_0 = 58$ Å (*Figure 2—figure supplement 5*).

The initial condition in the unsupervised learning is another issue. Since MSM has a larger number of parameters than in typical hidden Markov modeling, unsupervised learning requires a good initial condition for optimization to avoid being trapped in a local minimum. By using $\mathbf{T}_{\text{simulation}}(\tau)$ as the initial condition, we achieved a likelihood convergence with $lnL(\mathbf{T}(\tau)) = -584{,}947$ with 10,000-step optimization (taking one week using the parallel implementation of the Baum-Welch algorithm

[*Rabiner and Juang, 1986*]). For comparison, we also performed the optimization using a random matrix as the initial condition (*Figure 2—figure supplement 8C*). In the figure, the optimization of the likelihood looks stacked after 10,000 iterations and its value is lower than that of $\mathbf{T}_{\text{simulation}}(\tau)$ as the initial condition. This result suggests that the model from a random matrix could be ruled out because of trapping in a local minimum, or at least that global optimization of a random matrix is practically very inefficient. This also emphasizes the importance of transferring knowledge $\mathbf{T}_{\text{simulation}}(\tau)$ learned from simulations for improving the unsupervised learning $\mathbf{T}_{\text{experiment}}(\tau)$ from experimental data (*Torrey and Jude, 2009*). Although just $\mathbf{T}_{\text{simulation}}(\tau)$ was used as the initial condition here, advanced algorithms in transfer learning (*Torrey and Jude, 2009*) can be incorporated in a future study.

We here used the conventional constant-time binning for photon counting because the standard MSM is based on constant-time binning. A promising possibility for future studies is to apply continuous-time Markov modeling (*McGibbon and Pande, 2015*), which may allow us to use information-based binning (*Watkins and Yang, 2004*) or photon-by-photon analysis (*Gopich and Szabo, 2009*; *Okamoto and Sako, 2012*; *Pirchi et al., 2016*), avoiding photon counting noise.

In conclusion, exploiting the temporal information embedded in experimental *time-series* data to improve the simulation-based model has provided a rich, dynamic and experimentally consistent picture of the folding mechanism for the FBP WW domain. The data-assimilated MSM pathway could be used to improve the force-field parameters of proteins, nucleic acids, and other biomolecules. The semi-supervised learning combined with MSM method developed here is a quite general framework that can be used to understand conformational transitions in proteins and other biomolecules. It can be extended to interpret other experimental data possibly using more advanced techniques.

## Methods

### Molecular dynamics simulation

Monte Carlo searches were performed for labeling the dyes without any steric crashes with the protein. The constructed dye-labeled WW domain was solvated by TIP3P water molecules in a cubic box of 64.3 Å side length. Sodium ions were added to make the net charge of the system neutral.

In order to obtain unfolded structures as the initial structures for production runs, we first performed eleven 80 ns simulations at high temperature (600 K) in the NVT ensemble. Then, each system was equilibrated by 40 ns simulation in the NPT ensemble (1 atm and 370 K, slightly lower than the estimated melting temperature in the previous simulation with a different force field [*Mu et al., 2006*]). After determining the average volume size in these eleven trajectories, the volume size of each simulation was reset to the average value. Then, we conducted production simulation of eleven systems for 25.6 µs in the NVT ensemble (370 K). Furthermore, we performed six additional production runs of lengths 10 µs and another four simulations of 14 µs. All of these additional ten simulations started from the native structure.

All production simulations were conducted with the Amber 14 GPU version of the PMEMD module (*Salomon-Ferrer et al., 2013*) (using the SPFP precision model [*Le Grand et al., 2013*]) on GPU computers. Amber ff99SB (*Hornak et al., 2006*) was used for the force field. For the FRET dyes (Alexa 488, Alexa 594, and linkers), we used the AMBER-DYE force field (*Graen et al., 2014*), which is optimized for use with the Amber ff99SB and TIP3P water model. A cutoff of 8 Å was applied for the Lennard-Jones and short-range electrostatic interactions. For the long-range electrostatic interactions, we used the Particle Mesh Ewald method (*Darden et al., 1993*). All bonds involving hydrogen atoms were constrained with the SHAKE/SETTLE algorithm (*Miyamoto and Kollman, 1992*; *Ryckaert et al., 1977*). Using hydrogen mass repartitioning (*Hopkins et al., 2015*), a time step of 4 fs was used. Temperature and pressure were controlled by a Berendsen thermostat (*Berendsen et al., 1984*) with a coupling constant of 1 ps and the Monte Carlo barostat, respectively. Trajectories were saved every 200 ps. Q was calculated following the definition of *Best et al. (2013)*.

It is known that conventional force fields including Amber99SB used in this study overstabilize compact states in the unfolded or disordered states (*Piana et al., 2014, 2015*). Recently, Best and coworkers modified short-range proteinrwater pair interactions to correct this bias for a derivative of the Amber ff03 force field with the TIP4P/2005 water model (*Best et al., 2014*). Specifically, they

scaled the Lennard-Jones $\varepsilon_{Oi}$ between the oxygen of water molecules and all protein atoms by using a factor of 1.1. In order to sample non-compact conformations, we scaled $\varepsilon_{Oi}$ of Amber ff99SB and TIP3P in the same manner, and conducted folding simulations. Starting from the unfolded structures, which were generated in the NVT ensemble (600 K), we performed ten 7 μs simulations in the NPT ensemble (1 atm, and 360 K, slightly lower than the previous case for more conservative simulations). We confirmed that the unfolded states in these trajectories prefer more elongated conformations compared with the original Amber ff99SB. However, we did not observe any folding events, suggesting that the scaling may destabilize the native state at least in the case of Amber ff99SB and TIP3P (*Figure 2—figure supplement 1B*). Thus, we decided to use only the simulation data of the original Amber ff99SB in this work. For these additional simulations, we used GENESIS (*Jung et al., 2015*; *Kobayashi et al., 2017*) and K computer as well as the Amber 14 PMEMD module and GPU computers. All structural figures were prepared with VMD (*Humphrey et al., 1996*).

## Markov state model and semi-supervised learning

The regular spatial clustering was applied with RegularSpatial function in MSMBuilder (*Harrigan et al., 2017*) in the 2-D space spanned by $Q$ and $\varepsilon$.

$h(o_i|s_i)$ in the likelihood function (*Equation 2*) is the probability of observing donor and acceptor photon-counts $o_i = \{N_D, N_A\}$ given MSM's state $s_i$. Denoting the donor and acceptor photon count rates in the state $s_i$ by $n_D(s_i)$ and $n_A(s_i)$, this probability is given by the product of Poisson distributions (*Gopich and Szabo, 2012*),

$$h(o_i|s_i) = \frac{(n_D(s_i)\tau)^{N_D}}{N_D!} e^{-n_D(s_i)\tau} \frac{(n_A(s_i)\tau)^{N_A}}{N_A!} e^{-n_A(s_i)\tau}. \tag{3}$$

Following the previous analysis by Chung and coworkers (*Chung et al., 2012*), we applied the condition that the sum of the donor and acceptor count rates is independent of the conformational states, that is, $n = n_D(s_i) + n_A(s_i) \equiv \mathrm{const}$. This condition is met when the gamma factor, which is the ratio of the quantum yields and detection efficiencies of the acceptor and donor photons, is equal to one in all conformational states. Under this condition, *Equation 3* is rewritten as

$$h(o_i|s_i) = \left[ \frac{(n\tau)^{N_D+N_A}}{(N_D+N_A)!} e^{-n\tau} \right] \left[ \frac{(N_D+N_A)!}{N_D!N_A!} \varepsilon^{N_A} (1-\varepsilon)^{N_D} \right], \tag{4}$$

Here, the expected FRET efficiency $\varepsilon = n_A(s_i)/(n_D(s_i) + n_A(s_i))$ is related to the distance between donor and acceptor $r(s_i)$ through the Förster theory,

$$\varepsilon = \frac{1}{1 + [r(s_i)/R_0(\kappa^2)]^6}, \tag{5}$$

where $R_0$ and $\kappa^2$ are the Förster radius and the orientation factor between the transition dipoles of dyes, respectively. By analyzing the structures in MD simulations, we evaluated the contribution of the orientation factor $\kappa^2$ to $R_0$. We calculated the directions of the dipoles by assuming that the transition dipole moments are aligned with the long axis of each chromophore (*Best et al., 2015*). Rather than evaluating the orientation factor of each MSM state, we evaluated the averages and standard deviations of the orientation factor in four local regions defined along the $Q$ and $\varepsilon$ axes respectively because fluctuations in the instantaneous orientation factor required a large number of samples. The average (standard deviation) of each region along the $Q$ axis was $\kappa^2 = 0.63$ (0.64) for $Q = 0.00$–0.25, 0.63 (0.64), for $Q = 0.25$–0.50, 0.64 (0.63), for $Q = 0.50$–0.75, and 0.60 (0.62) for $Q = 0.75$–1.00. Also, the average (standard deviation) of each region along the $\varepsilon$ axis was $\kappa^2 = 0.66$ (0.69) for $\varepsilon = 0.00$–0.25, 0.64 (0.68) for $\varepsilon = 0.25$–0.50, 0.61 (0.66) for $\varepsilon = 0.50$–0.75, and 0.63 (0.64) for $\varepsilon = 0.75$–1.00. The results suggest that $\kappa^2$ hardly depends on states within standard deviations. Thus, we here employed the isotropic average approximation $\kappa^2 = 2/3$, and $R_0 = 56$ Å (*Jäger et al., 2006*) was used. In the same way, the donor-acceptor distance $r$ was calculated from the geometric centers of the donor and acceptor chromophores. The averaged value of $r$ within each state $s_i$ was used for $r(s_i)$.

The total log likelihood function $\ln L(\mathbf{T}(\tau)) = \sum_k \ln L_k(\mathbf{T}(\tau))$ of observing smFRET time-series data was optimized using the Baum-Welch algorithm (*Rabiner and Juang, 1986*), imposing the detailed-

balance condition as a constraint (*McGibbon et al., 2014a*; *Noé et al., 2013*). A numerical benefit of imposing the condition is that the maximum eigenvalue of the transition probability matrix always becomes one and its corresponding eigenvector represents the equilibrium probabilities of states. For this intensive calculation, in-house MATLAB codes (https://github.com/ymatsunaga/mdtoolbox) were developed and parallelized over photon-sequences. The codes are publicly available at https://github.com/ymatsunaga/mdtoolbox under the BSD 3-Clause License (*Matsunaga, 2018*); copy archived at https://github.com/elifesciences-publications/mdtoolbox). In the Baum-Welch algorithm, the parameters whose initial values are zero are always kept as zero. In order to relax this topological constraint, very weak random noise was added to $\mathbf{T}_{\text{simulation}}(\tau)$ before the optimization. In the early phase of the optimization (100 steps), unfolded states are stabilized irrespective of their compactness (with the likelihood value of $\ln L(\mathbf{T}(\tau)) = -588,314$, *Figure 2—figure supplement 7B*). Then, during the convergence of the likelihood in an optimization of 10,000 steps, the compact unfolded state disappeared while the folded state becomes stabilized (with a larger likelihood value of $\ln L(\mathbf{T}(\tau)) = -584,947$, *Figure 2—figure supplement 7C*).

In order to examine the overfitting of the model to the smFRET data, we divided smFRET data into halves, and the likelihood optimization was independently applied to the two subsets. Both results generated qualitatively similar network structures as with the full data set (*Figure 2—figure supplement 4*).

To see the dependence on the choice of the Förster radius $R_0$, we carried out the unsupervised learning using a set of different $R_0$ values ($R_0$ = 54 Å, 55 Å, 57 Å, and 58 Å) with the same 87 states. The overall structure of the MSM network was robust against the choice of $R_0$ except for $R_0$ = 58 Å (*Figure 2—figure supplement 5*).

It is known that *Equation (5)* only approximately holds at the weak excitation-limit of the donor dye. Here, its validity of the assumption is questionable because a very high intensity laser (10 kW/cm$^2$) was used in the current smFRET measurement to increase the number of photons (*Chung et al., 2012*). Thus, we examined the FRET efficiency outside the weak-excitation limit by using the following relation given by *Camley et al. (2009)*:

$$\varepsilon' = \frac{1}{\Lambda + [r(s_i)/R_0(\kappa^2)]^6}. \tag{6}$$

Here, $\Lambda$ depends on all rates of dye photophysics other than the energy transfer. $\Lambda = 1$ corresponds to the weak-field limit (*Equation 5*). $\Lambda > 1$ reflects the inability of doubly excited dye pairs to undergo FRET within commonly accepted physical models. Here, using the same 87 states, we optimized the likelihood function by using $\varepsilon'$ with $\Lambda = 1.065$, a value used in *Camley et al., 2009*. The optimized model is plotted in *Figure 2—figure supplement 6*. In the figure, although the intermediate states look more stabilized, the locations of stabilized states are qualitatively the same as the weak-field limit ($\Lambda = 1$, *Equation 5*). This suggests that the folding mechanism is robust against the definition of $\varepsilon$.

In order to evaluate the dependence on the initial condition, we also performed the optimization using a random matrix as the initial condition. The convergence of the likelihood function is shown in *Figure 2—figure supplement 8*.

## Analysis of Markov state models

We analyzed the dynamic properties of the constructed MSMs by generating long MSM simulation trajectories with stochastic simulations. We first generated trajectories of states by using $\mathbf{T}(\tau)$ ($\tau$ = 200 ns). Specifically, a random number between 0 and 1 was drawn at every step to determine which state the system will jump to in the next step according to $\mathbf{T}(\tau)$. For the reproducibility test against the original smFRET data, we generated a total of 10 independent trajectories of states each having the same time-length as the smFRET data, and then virtually emitted photons according to the likelihood function (*Equation 2*) from the states. We compared the histogram of observed FRET efficiencies $E = N_A/(N_A + N_D)$ using 200 ns and 50 µs time-windows.

In order to examine the overfitting again, we performed a *k*-fold cross validation test (with *k*=4) and calculated errors by using histograms of measured FRET efficiencies $E$. We partitioned the smFRET data set into *k* subsets (*k*-1 subsets as the training data, a single subset as the test data). We evaluated the mean squared error between the normalized FRET efficiency histograms

calculated from the model and the subsets of the smFRET data (shown in *Figure 3—figure supplement 1*). The prediction error (the so-called cross validation error) for the test data was found to be $11.8 \times 10^{-5}$, which is quite small. This suggests that overfitting is not a critical issue in our modeling.

Trajectories of conformations were generated from the trajectories of states by choosing a random conformation from a state at each step. These conformational trajectories were used to characterize the time-course behavior of $Q$ and the gyration radius (*Figure 4—figure supplement 1*), as well as the transition state ensemble (*Figure 4E and F*).

The folding behavior was further characterized by calculating $p_{fold}$, the probability of a given state to fold before it unfolds. The $p_{fold}$ was solved by applying the transition-path theory (*Metzner et al., 2009*; *Noé et al., 2009*) (with committors function in MSMBuilder [*Harrigan et al., 2017*]). The $p_{fold}$ was mapped onto geometric space (C$\alpha$-RMSDs of the hairpins 1 and 2) using the trajectories generated as described above.

We conducted pathway analysis from the unfolded to the folded state by decomposing the flux of folding trajectories into individual pathways (*Metzner et al., 2009*; *Noé et al., 2009*). In the algorithm, after calculating the net flux matrix between states, the largest flux pathway from the unfolded to the folded state was searched by using Dijkstra's algorithm. Then, the largest flux was subtracted from the net flux matrix. Subsequently, the second largest flux pathway was determined by using Dijkstra's algorithm. Representative pathways were obtained by repeating this procedure using the paths function of MSMbuilder (*Harrigan et al., 2017*).

## Acknowledgements

We extend our sincere thanks to William A Eaton and Hoi Sung Chung for providing the single-molecule FRET data and helpful comments on the draft. Thanks also to Satoshi Takahashi, Hiroyuki Oikawa, Simon Olsson and Suyong Re for their comments. We acknowledge help from Timo Graen in using the AMBER-DYES force field. Computational resources were provided by HOKUSAI GreatWave in the RIKEN Advanced Center for Computing and Communication and by K computer in the RIKEN Advanced Institute for Computational Science through the HPCI System Research project (Project IDs. hp160022, hp170137 and ra000009). This research has been funded as a RIKEN pioneering project 'Dynamic Structural Biology' (funding awarded to YS), as part of the strategic programs for innovation research ('Computational life science and application in drug discovery and medical development' and 'Novel measurement techniques for visualizing live protein molecules at work' [Grant No. 26119006] [to YS]), by JST CREST under the 'Structural Life Science and Advanced Core Technologies for Innovative Life Science Research' program (Grant No. JPMJCR13M3) (to YS), by MEXT Japan under their 'Initiative for High-Dimensional Data-Driven Science through Deepening of Sparse Modeling' (Grant No. 26120533) (to YM), and by JST PRESTO (Grant No. JPMJPR1679) (to YM).

## Additional information

### Funding

| Funder | Grant reference number | Author |
| --- | --- | --- |
| Japan Science and Technology Agency | JPMJPR1679 | Yasuhiro Matsunaga |
| Ministry of Education, Culture, Sports, Science, and Technology | 26120533 | Yasuhiro Matsunaga |
| RIKEN | Dynamic Structural Biology | Yuji Sugita |
| Research Organization for Information Science and Technology | hp160022 | Yasuhiro Matsunaga Yuji Sugita |
| Japan Science and Technology Agency | JPMJCR13M3 | Yuji Sugita |

| Ministry of Education, Culture, Sports, Science, and Technology | 26119006 | Yuji Sugita |
|---|---|---|
| Research Organization for Information Science and Technology | ra000009 | Yasuhiro Matsunaga<br>Yuji Sugita |

The funders had no role in study design, data collection and interpretation, or the decision to submit the work for publication.

## Author contributions

Yasuhiro Matsunaga, Conceptualization, Data curation, Software, Formal analysis, Funding acquisition, Validation, Investigation, Visualization, Methodology, Writing—original draft, Writing—review and editing; Yuji Sugita, Conceptualization, Data curation, Supervision, Funding acquisition, Writing—original draft, Project administration, Writing—review and editing

## Author ORCIDs

Yasuhiro Matsunaga ORCID http://orcid.org/0000-0003-2872-3908
Yuji Sugita ORCID http://orcid.org/0000-0001-9738-9216

## Decision letter and Author response

Decision letter https://doi.org/10.7554/eLife.32668.021
Author response https://doi.org/10.7554/eLife.32668.022

# Additional files

## Supplementary files

• Transparent reporting form
DOI: https://doi.org/10.7554/eLife.32668.019

## Data availability

All of the in-house program codes for the method proposed in this study, including test datasets, are available at GitHub: https://github.com/ymatsunaga/mdtoolbox.

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
