## [Decision Letter]

Thank you for submitting your article "Linking time series of single-molecule experiments with molecular dynamics simulations by machine learning" for consideration by *eLife*. Your article has been reviewed by three peer reviewers, and the evaluation has been overseen by a Reviewing Editor and Arup Chakraborty as the Senior Editor. The following individual involved in review of your submission has agreed to reveal his identity: John Straub (Reviewer #2).

The reviewers have discussed the reviews with one another and the Reviewing Editor has drafted this decision to help you prepare a revised submission.

Summary:

The manuscript describes a novel approach based on hidden Markov models to connect single-molecular FRET measurements with MD simulations, and application to the folding of formin-binding protein WW domain. The manuscript address two key issues in improving the validity of Markov state models used in protein folding studies, namely, 1) the construction of the discrete states, and 2) the construction of the continuous time Markov transition matrix T(\tau). Initially, discrete conformational states were obtained by clustering trajectory snapshots, and transition rates were then estimated from trajectories. To correct potential bias due to problems in the force field, the authors then used a hidden Markov model to model smFRET time series measurement and a) modified the discrete states and b) estimated the T(\tau) values between the discrete states so the smFRET is better explained. Indeed, the refined Markov state model can better account for the single-molecule time-series data: The improvement in reproducing the smFRET histogram of 50 uS as shown in Figure 3 is dramatic. It further revealed a different folding pathway of the WW-domain, as well as a different transition state ensemble.

The approach of using MD-derived MSM as basis and initial input for likelihood optimization with respect to smFRET data is very interesting and novel, and the calculations and uncertainty analysis were conducted carefully. The insight into the folding mechanism is very interesting. However, quite a few outstanding issues, including some serious concerns, remain to be addressed to assess whether publication is warranted.

Essential revisions:

1) An overall question about the current practice of Markov state models in MD-based protein folding studies. Assuming both simulation and experimental studies are under the same conditions (salt, dye, temperature, etc.), they are probing the same underlying physical process. Therefore, they should reflect the same ground-truth reality. As MD can provide atomistic details within the time scale of simulation, MD studies should provide far more details than smFRET. However, this is not the case. Rather, MD simulation cannot stand on its own, and needs to be corrected by smFRET. As described by the authors, the MSM is shifted before and after incorporating smFRET data. The authors showed that the discrete states are very different when smFRET data are brought into consideration. While it is not explicitly stated (please clarify), it seems that the refined Markov states are derived from the original Markov states by clustering of MD trajectories, but updated with new equilibrium probability (hence different bubble areas in Figure 2) and new transition rates. How can the ground truth, for example, equilibrium probabilities of the Markov states, change if it is examined one way (by MD) or another way (MD+smFRET)? This would suggest there is no objective ground truth that is accessible by current MD simulations.

Lending support to this conclusion, is the discussion you provided, that in other studies prior to this manuscript, the number of discrete states and the nature of these states are all malleable, depending on which metric one examines (RMSD or contact map). This would again suggest that there is no objective truth in determining what constitutes a set of discrete states that are Markovian and what their transition rates are. Rather, all depends on which metric and/or which additional experimental data one chooses to examine.

One cannot help but speculate: would one expect some other alternative discrete Markovian states different from the currently reported ones to emerge, when other types of experimental time series data other than \epsilon is incorporated. Is it likely that yet more different Markovian states, different dynamics of transition rates, different folding dynamics, and different transition state ensembles will be identified, all for the same WW-domain under the same condition (dye, pH, salt, temperature, if well controlled)?

This aspect of shifting ground truth in MD simulation and Markov state modeling is rather unsettling. A pessimistic, but not unreasonable, view is that without a principled approach in defining true reaction coordinates for protein folding one has to make do with rather ad hoc and heuristic approaches in defining the discrete states with unexamined consequences, and hence this unsettling aspect may be with us for a long while and the truth may be elusive.

Here is a suggestion in this respect: The authors employ Equation 5 for a model of FRET efficiency, however, more detailed models exist such as http://aip.scitation.org/doi/pdf/10.1063/1.3230974

The authors could evaluate their approach and determine if a more detailed model might be justified or needed in making contact with experimental measurements.

2) While concern 1 is more general, there is another question more specific to connecting MD and the smFRET measurements. It will be necessary for the authors to show how well the HMM learned T(\tau) conform with MD simulation generated T(\tau) after the refined discrete states are obtained, in a statistically significant way. For example, the authors should compare the HMM rates and the MD rates for the subsets of state-state transitions with the highest and relevant lowest rates among the 87x87 parameters. In addition, the flux and pathway analysis should also be carried out using MD-derived rates after the refined states are obtained by the HMM model, for key paths carrying most of the flux, and compared with the analysis using HMM rates. The authors may have already done some of these in the transition state ensemble analysis, and may already have all necessary data.

This is important for validating the connection between the sm-FRET (HMM modeling) time series and the MD trajectories the authors are proposing. It would be fantastic if all works out, as this would naturally suggest that in the future one no longer needs to do MD simulation. Rather, one could use smFRET to fit dynamic rates, as long as the discrete states can be obtained, for example, by some other means. However, one should be cautious about being overly optimistic here, as there are significant problems in the enormous search space of very high dimension. It will be very interesting if the authors can show that the HMM rates and MD rates after refinement are the same.

3) Potential over-fitting remains a concern, as there is a large number (87x87 = 7,569) parameters that need to be fitted. While the authors qualitatively compared results using each half of the data through plotting, it would be necessary to do some cross-validation tests, which are standard in machine learning. Specifically, the authors could use (n-1)/n balanced smFRET and MD data (e.g. similar number of folding trajectories) to identify refined states, and estimating \tau values, then test whether the remaining 1/n data falls within these defined states, and whether the unseen epsilon histograms can be reproduced accurately.

4) In Equation 5, the average values are used in calculating \epsilon. It seems that kappa has little difference at different regions of Q. Is the difference along the dimension of \epsilon that is used for clustering due to difference in distances *r(s_i_*)? It will be helpful to also show both mean and variance of *r(s_i_*) for each of the major discrete states, and justify whether the degree of in-cluster and between-cluster heterogeneity/homogeneity conforms to the Markovian assumption of each discrete state. This relates to the issue of validating the defined discrete states.

5) The calculations of FRET efficiencies from MD trajectories need to be stated explicitly. Furthermore, the estimation of FRET efficiencies from experimentally measured photon sequences also needs to be presently clearly. The latter in fact is a long-standing issue in the single-molecule community, for example, the time binning convention and the information-base binning by Haw Yang.

6) It is not obvious how the folding pathways were decomposed into individual pathways for both MSM models. How this procedure was conducted would critically impact the interpretation of results. This part thus needs careful and clear explanation.

7) Certain key references of interring dynamics from smFRET trajectories were missed in the context of this work, such as Haas et al., 2013.

8) Obtaining a solution of similar likelihood value starting from a random matrix is a bit worrisome. Can the authors elaborate more on how this model was ruled out? Furthermore, the convergence of likelihood optimization should also be discussed to reveal the robustness of this approach.

[Editors' note: further revisions were requested prior to acceptance, as described below.]

Thank you for resubmitting your work entitled "Linking time series of single-molecule experiments with molecular dynamics simulations by machine learning" for further consideration at *eLife*. Your revised article has been favorably evaluated by Arup Chakraborty (Senior Editor), a Reviewing Editor, and three reviewers.

The manuscript has been improved but there are some remaining issues that need to be addressed before acceptance, as outlined below:

Overview:

The unsettling aspects of shifting ground truth, or rather, the lack of ground truth remains. This is, however, an issue the community of protein folding simulations has to deal with.

Using *Q* as an adequate reaction coordinate is similarly problematic and inadequate, as *Q* is no substitute to reaction coordinates in the strict sense of earlier classic studies of Chandler and Dinner. But this again is a community problem, and we do not wish to specifically penalize the authors of the current work. Therefore, we accept the authors' arguments, although at least one of the reviewers had some fundamental concerns about the entire set of approaches in this area.

Specific points to address:

1) There is one issue that the authors did not provide answers to but they can easily do, namely, regarding the question of providing details of the 87x87 rates from the Hidden Markov model and the corresponding rates from MD simulation. It will be necessary to make first a general statement on how similar/different these two sets of matched/paired rates are, and second, provide details, e.g., in the Appendix, with side-by-side or overlaid histograms. If these two distributions are similar, it would be a comforting result. However, even if these two distributions do not agree, it will be important for readers to know this fact so they can directly draw their own inferences. This can be easily done with existing data.

2) A grammar and expression overhaul is highly recommended to enhance the readability of the work.

---

## [Author Response]

Essential revisions:1) An overall question about the current practice of Markov state models in MD-based protein folding studies. Assuming both simulation and experimental studies are under the same conditions (salt, dye, temperature, etc.), they are probing the same underlying physical process. Therefore, they should reflect the same ground-truth reality. As MD can provide atomistic details within the time scale of simulation, MD studies should provide far more details than smFRET. However, this is not the case. Rather, MD simulation cannot stand on its own, and needs to be corrected by smFRET. As described by the authors, the MSM is shifted before and after incorporating smFRET data. The authors showed that the discrete states are very different when smFRET data are brought into consideration. While it is not explicitly stated (please clarify), it seems that the refined Markov states are derived from the original Markov states by clustering of MD trajectories, but updated with new equilibrium probability (hence different bubble areas in Figure 2) and new transition rates. How can the ground truth, for example, equilibrium probabilities of the Markov states, change if it is examined one way (by MD) or another way (MD+smFRET)? This would suggest there is no objective ground truth that is accessible by current MD simulations.

We agree with the reviewers that both simulation and experimental data should reflect the same ground-truth reality. However, despite improvements over the decades, force field parameters used in MD simulation are not perfectly correct as reaching the ground-truth reality. In particular, while the local interactions are well described by the current force fields it is still difficult to reproduce the energetic balance between global states (such as unfolded and folded states). Indeed, Piana and coworkers (Piana et al., 2011) showed, in their protein folding simulation study, the folding mechanism of the villin headpiece depends substantially on the choice of force field. It is also known that most force fields produce unfolded states that are more compact and structured than those suggested experimentally (Piana et al., 2014). As suggested by these studies, a subtle free energy balance between the unfolded and folded states is crucial for determining the folding mechanism. Considering this background, the motivation of this work is to make MD data toward the ground-truth reality in a data-driven way. We reweighted the transition probabilities by using smFRET data to achieve the subtle balance in free energy. In the future study, using the free energy landscape and folding pathways obtained by our scheme, it is possible to improve force field parameters closer to the ground truth reality. According to this discussion, we have revised the Introduction section (second paragraph).

Lending support to this conclusion, is the discussion you provided, that in other studies prior to this manuscript, the number of discrete states and the nature of these states are all malleable, depending on which metric one examines (RMSD or contact map). This would again suggest that there is no objective truth in determining what constitutes a set of discrete states that are Markovian and what their transition rates are. Rather, all depends on which metric and/or which additional experimental data one chooses to examine.One cannot help but speculate: would one expect some other alternative discrete Markovian states different from the currently reported ones to emerge, when other types of experimental time series data other than \epsilon is incorporated. Is it likely that yet more different Markovian states, different dynamics of transition rates, different folding dynamics, and different transition state ensembles will be identified, all for the same WW-domain under the same condition (dye, pH, salt, temperature, if well controlled)?This aspect of shifting ground truth in MD simulation and Markov state modeling is rather unsettling. A pessimistic, but not unreasonable, view is that without a principled approach in defining true reaction coordinates for protein folding one has to make do with rather ad hoc and heuristic approaches in defining the discrete states with unexamined consequences, and hence this unsettling aspect may be with us for a long while and the truth may be elusive.

We agree with the reviewers that Markov state modeling depends on the choice of coordinate or metric one examines. It generally quite difficult to find the true reaction coordinate for defining states in complex systems. On the other hand, in the case of protein folding problem, theoretical and simulation studies over the decades have shown that the fraction of native contacts, *Q*, is the best reaction coordinate for describing folding process. For example, Best and coworkers, by analyzing folding MD simulation data of various proteins, showed that *Q* successfully captures the transition states for all the proteins (Best, Hummer and Eaton, 2013). The FRET efficiency \epsilon was chosen for matching with smFRET data and for differentiating the compact and extended states. These two coordinates (*Q* and \epsilon) are necessary and sufficient for describing the progress of folding as well as refining the bias of the current force field parameters toward compact structures. Also, since the structure of WW domain is rather simple (three-strands, betasheet-only), these coordinates are enough for uniquely identifying tertiary structures. Considering these reasons and the validation through the comparison with the Phi-value analysis, we think that the picture obtained by the current MSM may be the robust against the choice of coordinates. We have added this discussion in the Results section (subsection “Molecular dynamics simulations”, second paragraph).

Here is a suggestion in this respect: The authors employ Equation 5 for a model of FRET efficiency, however, more detailed models exist such as http://aip.scitation.org/doi/pdf/10.1063/1.3230974The authors could evaluate their approach and determine if a more detailed model might be justified or needed in making contact with experimental measurements.

We thank the reviewers for letting us know the important paper. Following the reviewers’ suggestion, we applied our scheme with the new definition for FRE efficiency 𝜀 = 1⁄[Λ + (𝑟⁄𝑅_*_)^-^] where Λ > 1. Here, we optimized the likelihood function with Λ = 1.065, a value used in the paper. The result has been added as Figure 2—figure supplement 6. In the figure, although the transition states look more stabilized, the locations of stabilized states are qualitatively same as the original one (Λ = 1.0). This suggests that the story of the manuscript would be robust against the definition of ε (subsection “Markov state model and semi-supervised learning”, seventh paragraph).

2) While concern 1 is more general, there is another question more specific to connecting MD and the smFRET measurements. It will be necessary for the authors to show how well the HMM learned T(\tau) conform with MD simulation generated T(\tau) after the refined discrete states are obtained, in a statistically significant way. For example, the authors should compare the HMM rates and the MD rates for the subsets of state-state transitions with the highest and relevant lowest rates among the 87x87 parameters. In addition, the flux and pathway analysis should also be carried out using MD-derived rates after the refined states are obtained by the HMM model, for key paths carrying most of the flux, and compared with the analysis using HMM rates. The authors may have already done some of these in the transition state ensemble analysis, and may already have all necessary data.This is important for validating the connection between the sm-FRET (HMM modeling) time series and the MD trajectories the authors are proposing. It would be fantastic if all works out, as this would naturally suggest that in the future one no longer needs to do MD simulation. Rather, one could use smFRET to fit dynamic rates, as long as the discrete states can be obtained, for example, by some other means. However, one should be cautious about being overly optimistic here, as there are significant problems in the enormous search space of very high dimension. It will be very interesting if the authors can show that the HMM rates and MD rates after refinement are the same.

In this study, the states were defined through the structural clustering of 400 microseconds MD simulation data. As 400 microseconds should be long enough for sampling all possible structures of WW domain, we think that the defined states are transferable during the HMM refinement. We would like to point that we already characterized the difference in T(\tau) from the MD and HMM refinement: Figure 4 shows the result of flux analysis of folding trajectories, which reveals the bottleneck of folding (the transition state ensemble). Figure 5 shows the results of the pathway analysis, which reveals folding pathways with largest fluxes.

As the reviewers pointed out, it would be nice if all transition probabilities are accurately estimated by the HMM from smFRET data. However, as shown by the HMM refinement starting from a random matrix (in response to comment 8), it is still difficult to estimate accurate kinetics along the *Q* axis from the limited number of smFRET observations. This suggests an importance of complementary use of both data (MD and smFRET), as proposed by our scheme (the MD information as the initial condition for HMM).

3) Potential over-fitting remains a concern, as there is a large number (87x87 = 7,569) parameters that need to be fitted. While the authors qualitatively compared results using each half of the data through plotting, it would be necessary to do some cross-validation tests, which are standard in machine learning. Specifically, the authors could use (n-1)/n balanced smFRET and MD data (e.g. similar number of folding trajectories) to identify refined states, and estimating \tau values, then test whether the remaining 1/n data falls within these defined states, and whether the unseen epsilon histograms can be reproduced accurately.

We thank the reviewers for the important suggestion. Following the reviewers’ comment, we have performed a *k*-fold cross validation test (with *k*=4) by partitioning the smFRET data set into *k* subsets (k-1 subsets as the training data, a single subset as the test data). We have evaluated the mean squared errors between two normalized FRET efficiency histograms calculated from the model and the subsets of smFRET data. The prediction error (the so-called cross validation error) for the test data was found to be 11.8 × 10^78^, which is quite small. This suggests that the overfitting may not be a major issue in our modeling. We have added this result (Figure 3—figure supplement 1) in the Materials and methods section (subsection “Analysis of Markov state models”, second paragraph).

4) In Equation 5, the average values are used in calculating \epsilon. It seems that kappa has little difference at different regions of Q. Is the difference along the dimension of \epsilon that is used for clustering due to difference in distances r(s_i_)? It will be helpful to also show both mean and variance of r(s_i_) for each of the major discrete states, and justify whether the degree of in-cluster and between-cluster heterogeneity/homogeneity conforms to the Markovian assumption of each discrete state. This relates to the issue of validating the defined discrete states.

Following the reviewers’ comment, we have calculated the orientation factor k^2^ along the e (FRET efficiency) axis. The average (standard deviation) in each region was k^2^ = 0.66 (0.69) for e = 0.00-0.25, 0.64 (0.68) for e = 0.25-0.50, 0.61 (0.66) for e = 0.50-0.75, and 0.63 (0.64) for e = 0.75-1.00. Since the orientation factor k^2^ show little dependence on both *Q* and e axes, it would be reasonable to use the isotropic average approximation as employed in the manuscript. We have added this result in the Materials and methods section (subsection “Markov state model and semi-supervised learning”, third paragraph).

Also, we have calculated both mean and variance of donor-acceptor distance *r(s_i_*) for each state. It has been plotted in the 2D space spanned by *Q* and *r* (added as Figure 2—figure supplement 2). The figure shows that there are distance gaps between states, which is supportive for the Markovian assumption for the states. Large standard deviations are observed within states of small donor-acceptor distances. This is due to the worsening of spatial resolution in FRET efficiency when *r(s_i_*) is small compared to *R*_0_ as discussed in the information-binning study by Haw-Yang and coworkers (Watkins and Yang, 2004). We have added this result (Figure 2—figure supplement 2) in the Materials and methods section (subsection “Molecular dynamics simulations”, last paragraph).

5) The calculations of FRET efficiencies from MD trajectories need to be stated explicitly. Furthermore, the estimation of FRET efficiencies from experimentally measured photon sequences also needs to be presently clearly. The latter in fact is a long-standing issue in the single-molecule community, for example, the time binning convention and the information-base binning by Haw Yang.

We have explicitly described the calculations of FRET efficiencies from MD trajectories and photon sequences in the Results and Materials and methods sections. In particular, we have now used different terms for two FRET efficiencies (‘FRET efficiency \epsilon’ estimated from MD, ‘Measured FRET efficiency *E*’ from photon counts). We here used the conventional constant-time binning because the Markov state modeling is based on the constant-time binning. As a future direction, it is promising to apply a continuous-time Markov modeling (McGibbon and Pande, 2015), which may allow us to use the information-based binning (Watkins and Yang, 2004) or photon-by-photon analysis (Gopich and A. Szabo, 2009). We have described this direction in the Discussion section, adding citations for these works (fifth paragraph).

6) It is not obvious how the folding pathways were decomposed into individual pathways for both MSM models. How this procedure was conducted would critically impact the interpretation of results. This part thus needs careful and clear explanation.

We have described how the folding pathways were decomposed and its impact in the Results and Materials and methods sections. In the Results section we have added the following:

“The flux of folding trajectories was decomposed into individual pathways for both models (Figures 5A and 5B). […] The figures show dominant folding pathways with each of largest fluxes contributing to 50% of the total flux.”

7) Certain key references of interring dynamics from smFRET trajectories were missed in the context of this work, such as Haas et al., 2013.

We thank the reviewers for letting us know the important paper. We have added following references for recent advances in extracting more dynamical and structural information from smFRET trajectory (Introduction, first paragraph):

Haas, Yang and Chu, 2013; Matsunaga, Kidera, and Sugita, 2015; Sun, Morrell and Yang, 2016; Hoefling et al., 2011.

8) Obtaining a solution of similar likelihood value starting from a random matrix is a bit worrisome. Can the authors elaborate more on how this model was ruled out? Furthermore, the convergence of likelihood optimization should also be discussed to reveal the robustness of this approach.

We have extended the optimization calculation of a random matrix and examined the convergence of the log likelihood function. The result has been added as Figure 2—figure supplement 8. In the figure, the optimization of the log likelihood from random matrix looks stacked after 10000 iterations and its value is lower than that of MD data. This suggests that the optimization gets trapped in a local minimum in the parameter space. From these observations, we conclude that the model from a random matrix could be ruled out, or at least, global optimization of a random matrix is practically very inefficient. We have added this result (Figure 2—figure supplement 8) in the Materials and methods section (subsection “Markov state model and semi-supervised learning”, last paragraph).

[Editors' note: further revisions were requested prior to acceptance, as described below.]

The manuscript has been improved but there are some remaining issues that need to be addressed before acceptance, as outlined below:Overview:The unsettling aspects of shifting ground truth, or rather, the lack of ground truth remains. This is, however, an issue the community of protein folding simulations has to deal with.Using Q as an adequate reaction coordinate is similarly problematic and inadequate, as Q is no substitute to reaction coordinates in the strict sense of earlier classic studies of Chandler and Dinner. But this again is a community problem, and we do not wish to specifically penalize the authors of the current work. Therefore, we accept the authors' arguments, although at least one of the reviewers had some fundamental concerns about the entire set of approaches in this area.Specific points to address:1) There is one issue that the authors did not provide answers to but they can easily do, namely, regarding the question of providing details of the 87x87 rates from the Hidden Markov model and the corresponding rates from MD simulation. It will be necessary to make first a general statement on how similar/different these two sets of matched/paired rates are, and second, provide details, e.g., in the Appendix, with side-by-side or overlaid histograms. If these two distributions are similar, it would be a comforting result. However, even if these two distributions do not agree, it will be important for readers to know this fact so they can directly draw their own inferences. This can be easily done with existing data.

Following the reviewers’ comment, we have added new plots comparing the rates of the two MSMs as Figure 2—figure supplement 4.

First (in panel A), we decomposed each transition probability matrix to obtain characteristic time scales before and after the hidden Markov modeling. We can see that the slowest time scale increases from 2.6 microseconds to 100 microseconds after the hidden Markov modeling. These time scales are related to folding/unfolding transitions of WW domain. Generally, simulation time scale is faster than experiments because of lower viscosity of TIP3P water model. This gap was improved by the information of the single-molecule experimental data.

In panel B, 87x87 *T_ij_* rates before and after the hidden Markov modeling are directly compared in a scatter plot where each rate is colored using the FRET efficiency e_i_. *T_ij_* rates from states with high FRET efficiencies are well correlated with each other. This means that *T_ij_* rates between compact states derived from MD simulations are consistent with the experimental data. In contrast, *T_ij_* rates from states with middle or low FRET efficiencies are less correlated. This means that hidden Markov modeling mainly “refined” the rates between elongated states.

In this revision, we have added these explanations in the main text.

2) A grammar and expression overhaul is highly recommended to enhance the readability of the work.

In this revision, a native speaker has carefully edited the language. We have also checked a number of expressions.